# Intrapartum Cesarean Section and Perinatal Outcomes after Epidural Analgesia or Remifentanil-PCA in Breech and Twin Deliveries

**DOI:** 10.3390/medicina59061026

**Published:** 2023-05-25

**Authors:** Miha Lucovnik, Ivan Verdenik, Tatjana Stopar Pintaric

**Affiliations:** 1Department of Perinatology, Division of Obstetrics and Gynaecology, University Medical Centre Ljubljana, Slajmerjeva 3, 1000 Ljubljana, Slovenia; ivan.verdenik@guest.arnes.si; 2Medical Faculty, University of Ljubljana, Vrazov trg 2, 1000 Ljubljana, Slovenia; 3Department of Anaesthesiology and Intensive Therapy, University Medical Centre Ljubljana, Slajmerjeva 3, 1000 Ljubljana, Slovenia; tatjana.stopar@kclj.si; 4Institute of Anatomy, Medical Faculty, University of Ljubljana, Vrazov trg 2, 1000 Ljubljana, Slovenia

**Keywords:** labor analgesia, epidural analgesia, remifentanil patient-controlled analgesia, breech delivery, twin delivery, cesarean section, labor outcomes

## Abstract

Comparative data on the potential impact of various forms of labor analgesia on the mode of delivery and neonatal complications in vaginal deliveries of singleton breech and twin fetuses are lacking. The present study aimed to determine the associations between type of labor analgesia (epidural analgesia (EA) vs. remifentanil patient-controlled analgesia (PCA)) and intrapartum cesarean sections (CS), and maternal and neonatal adverse outcomes in breech and twin vaginal births. A retrospective analysis of planned vaginal breech and twin deliveries at the Department of Perinatology, University Medical Centre Ljubljana, was performed for the period 2013–2021, using data obtained from the Slovenian National Perinatal Information System. The pre-specified outcomes studied were the rates of CS in labor, postpartum hemorrhage, obstetric anal sphincter injury (OASI), an Apgar score of <7 at 5 min after birth, birth asphyxia, and neonatal intensive care admission. A total of 371 deliveries were analyzed, including 127 term breech and 244 twin births. There were no statistically significant nor clinically relevant differences between the EA and remifentanil-PCA groups in any of the outcomes studied. Our findings suggest that both EA and remifentanil-PCA are safe and comparable in terms of labor outcomes in singleton breech and twin deliveries.

## 1. Introduction

Epidural analgesia (EA) is generally considered the gold standard for pain relief during labor and delivery. However, despite the current endorsement in major obstetric anesthesia guidelines [1,2] and extensive clinical use, many questions remain about the effects of EA on maternal and neonatal outcomes (mode of delivery, progress of labor, pain relief, perinatal hypoxia-ischemia, overall labor experience, and long-term outcomes). While previous randomized controlled trials showed lower maternal pain scores and higher maternal satisfaction with an epidural compared to other analgesic techniques used during labor [3,4,5,6], a recent Cochrane systematic review rated the evidence supporting the superiority of EA in these outcomes as low-quality [7]. The study concluded that compared to non-epidural methods of labor analgesia, EA did not affect the risks of cesarean delivery and had no immediate impact on Apgar scores or neonatal intensive care admissions [7]. Although EA has been associated with a prolonged second stage of labor and an increased risk of instrumental vaginal delivery [7,8], several studies in the last decade have reported that it does not increase the overall rate of cesarean sections (CS) [9,10,11,12]. Nevertheless, other uncertainties persist regarding the role of obstetric EA, including its routine use in labor, the timing of initiation, its impact on fetal positioning, and its role in high-risk or complicated deliveries [8,13,14,15,16]. Furthermore, characteristics such as multiple birth and fetal presentation may impact labor progress and maternal as well as neonatal outcome. These have not been adequately accounted for in randomized trials published to date, since smaller sub-groups (e.g., breech or twin labors) have not been sufficiently represented due to their relatively small contribution to the overall number of deliveries. As a result, it is not clear whether data on the impact of EA on labor and perinatal outcomes can be generalized to all sub-categories of laboring women (e.g., those with twins or breech fetuses).

Besides neuraxial analgesia, different pharmacological options are available for pain relief during labor, including parenteral opioids. Although it does not match labor EA’s analgesic efficacy, remifentanil patient-controlled analgesia (remifentanil-PCA) has emerged over the last two decades as an increasingly popular alternative method of labor analgesia as it provides acceptable maternal satisfaction ratings [17,18,19,20]. Remifentanil is a synthetic μ-opioid receptor agonist with an ultra-short duration of action. While its current use for obstetric analgesia is off-label and remains debatable, remifentanil-PCA is a clinically rational option in the setting of EA relative or absolute contraindication (e.g., prophylactic anticoagulant treatment, presence of coagulopathies, local infection, or maternal hypovolemia), EA unavailability, or the maternal preference for an EA alternative [19,21]. This analgesic modality is gaining increasing acceptance in many countries, although its reported association with respiratory depression [22,23,24] remains an obstacle to routine use in obstetric practice in many jurisdictions. A recent Cochrane review comparing remifentanil-PCA with alternative parenteral labor analgesic modalities noted that there is currently low-quality evidence to inform clinical practice regarding maternal and neonatal safety outcomes for remifentanil-PCA (maternal apnea and respiratory depression as well as neonatal Apgar score) [19]. Nevertheless, our group and others found no evidence of increased CS rate, serious maternal complications, operative vaginal delivery rate, non-reassuring fetal status during labor, or an Apgar score of less than 7 at five minutes from birth, with remifentanil-PCA compared to EA [25,26,27,28,29].

It is unclear if data on labor outcomes associated with EA and remifentanil-PCA can be generalized to all groups of women in labor. This is especially true for vaginal breech and twin deliveries, where associated problems such as prematurity, trauma, and hypoxia necessitate special anesthetic and obstetric considerations [30]. Both breech presentation and multiple gestations are associated with an increased risk of perinatal and neonatal morbidity and death, and the optimal delivery route is a critical but controversial aspect of intrapartum management [31,32,33]. When strict criteria are met before and during labor, planned vaginal delivery of singleton fetuses in breech presentation at term as well as twins with the first twin in cephalic presentation, remains a safe option [34,35,36]. Comparative data on the potential impact of various forms of labor analgesia on the mode of delivery and neonatal complications in vaginal deliveries of singleton breech fetuses and twins remain scarce [37,38,39,40]. Accordingly, the objective of the present study was to explore the associations between type of labor analgesia (EA vs. remifentanil-PCA) and CS in labor, and maternal outcomes (postpartum hemorrhage and perineal trauma) as well as neonatal outcomes (low Apgar score, birth asphyxia and neonatal admission to intensive care unit (NICU)) in planned breech and twin vaginal births.

## 2. Materials and Methods

We analyzed the Slovenian National Perinatal Information System (NPIS) data from 2013 to 2021. This retrospective study of anonymous entries was exempt from approval by the institutional ethical committee. NPIS records data from all deliveries in Slovenia at ≥22 weeks of pregnancy or when the birth weight is equal to 500 g or above since 1987. Registration is mandatory by law in the country’s 14 maternity units, and more than 140 variables are entered into a computerized database by the attending midwife and doctor.

We included planned vaginal breech and twin deliveries at the Department of Perinatology, University Medical Centre Ljubljana, during the study period. Our institution is Slovenia’s largest tertiary perinatal center, with almost 1/3 of all deliveries in the country (5000 to 5500 deliveries per year) and an overall CS rate of 21% [10,25]. When criteria are met, we routinely offer planned vaginal birth to women with breech singleton fetuses at term and twins [34,36]. The following criteria for the trial of vaginal breech delivery have been followed throughout the study period: no contraindication to vaginal birth (e.g., placenta previa); estimated fetal weight at least 2500 g and not more than 4000 g; no hyperextension of the fetal head (i.e., an extension angle of greater than 90 degrees); frank or complete breech presentation (incomplete of footling breech presentations were considered contraindications); adequate maternal pelvis; the absence of fetal anomaly that may cause dystocia; staff skilled in breech delivery and immediate availability of facilities for safe emergency cesarean delivery; and no cord presentation. In twins, vaginal delivery was recommended when the presenting twin was in cephalic presentation. Our planned CS rate for term breech deliveries is 60–70% for nulliparous and 30–40% for multiparous women, and 60–65% for twins (NPIS data). In the present study, we included women who received either remifentanil-PCA or EA at ≥37^+0^ weeks for singleton breech deliveries or twin deliveries at ≥34^+0^ weeks. Only women with spontaneous onset of labor or induction of labor were included in the analysis. Women with planned (pre-labor) CS were excluded from the analysis. Decision on the labor analgesia method (remifentanil-PCA or EA) was taken by women during labor after consulting with anesthesiologists, obstetricians and midwives.

Remifentanil-PCA has been used at our institution since 2013 per the standard operative protocol of the Department of Anaesthesiology and Intensive Therapy, University Medical Centre Ljubljana. Remifentanil hydrochloride (Ultiva, GlaxoSmithKline, Oslo, Norway) was diluted in normal saline to a concentration of 40 µg/mL and administered in the active phase of the first stage of labor from 20 to 40 µg with a bolus duration of 20 s, lockout interval of 2 min and no background infusion (Rhythmic™ Evolution, Micrel Medical Devices, Athens, Greece). Women were monitored for SpO_2_, heart rate, and etCO_2_ continuously (Capnostream^®^ capnography (Oridion^®^, Jerusalem, Israel)) and for blood pressure every 30 min. Parturients had obligatory one-to-one midwife care and supplemental oxygen (2 L/min) via a nasal catheter. Cardiotocography (CTG) was continuously monitored (Hewlett Packard Viridia Series 50IP^®^, Hewlett Packard, Palo Alto, CA, USA or Philips 50XM^®^, Amsterdam, The Netherlands). Remifentanil-PCA was stopped if pathological CTG changes occurred, including decreased variability, bradycardia, tachycardia, or late decelerations [18]. Per institutional protocol, contraindications for remifentanil-PCA in labor include the patient’s refusal, a history of opioid allergy, administration of parenteral opioids in the previous four hours, and non-availability of 1:1 midwife care.

During the study period, EA was provided with 0.1% bupivacaine and 2 µg of fentanyl per ml of local anesthetic by a combination of programmed intermittent epidural bolus (PIEB) and patient-controlled EA (PCEA) techniques using a pump (Rhythmic™ Evolution, Micrel Medical Devices, Athens, Greece). The epidural catheter was inserted by the attending obstetric anesthetist in accordance with the standard institutional protocols and following a strict aseptic technique. Briefly, with the parturient sitting, an epidural puncture was performed at either L2–L3 or L3–L4; the epidural space was located by loss-of-resistance to air, and a multi-orifice epidural catheter was inserted following the confirmation that there was no free cerebrospinal fluid present. Following insertion of the epidural catheter, an initial test dose of 10 mL of 0.1% bupivacaine plus fentanyl 2 μg/mL mixture was administered. The EA regimen of bupivacaine 0.1% with fentanyl 2 μg mL^−1^ was subsequently administered with an hourly PIEB of 5 mL (increased up to 10 mL if required) plus a PCEA regimen of 5 mL bolus with a 10 min lockout period. PCEA was commenced after satisfactory EA had been established, and the maternal hemodynamic status and the fetal heart rate were reassuring. According to the institutional protocol, the contraindications for EA include the patient’s refusal and presence of localized sepsis over the puncture site, thrombocytopenia, coagulopathy or anticoagulant treatment, hypovolemia or other evidence of cardiovascular instability, and severe lumbar deformity or previous major spinal surgery.

The pre-specified outcomes studied were the rates of CS in labor, postpartum hemorrhage (defined as >500 mL weighted blood loss and need for at least one unit of red-blood-cell transfusion), OASI (obstetric anal sphincter injury, i.e., 3rd- or 4th-degree perineal tears) rates, an Apgar score of <7 at 5 min after birth, birth asphyxia (diagnosed by the attending neonatologist and classified as mild-to-moderate (ICD-10 code P21.1) or severe (ICD-10 code P21.0)), and neonatal intensive care unit (NICU) admission. The second twin’s Apgar score of <7 at 5 min, birth asphyxia rates and NICU admission were analyzed for twin deliveries. We chose to focus on the second twin’s adverse neonatal outcomes, as labor complications after the birth of the first twin can potentially be attributed to the method of labor analgesia, and given the higher risk of adverse outcomes for the second twin reported in most observational studies to date [36,41,42,43,44]. Based on data from the previous study by Chadha et al. [39], a sample size of 103 breech deliveries (37 with EA and 67 controls) would be sufficient to reach a statistical power of 0.8 with an α level of 0.05 when analyzing intrapartum CS rates. For twins, a sample of 10 births with EA and 50 with remifentanil-PCA would be sufficient to reach the same statistical power for intrapartum CS analysis based on data from Farghali et al. [45]. Groups with remifentanil-PCA vs. EA were compared using the Chi-square test. *p* < 0.05 was considered statistically significant. The software used for statistical analysis was IBM SPSS Statistics for Windows Version 27.0 (Armonk, NY, USA: IBM Corp.).

## 3. Results

During the study period, there were 371 deliveries fulfilling the inclusion criteria; 127 term breech and 244 twin births. Table 1 presents demographic and obstetric characteristics of groups receiving EA and remifentanil-PCA in term breech deliveries. There were no significant differences between the groups regarding maternal age, body-mass index, nulliparity, assisted conception, mode of onset of labor, gestational age at delivery, fetal birth weight, and fetal size.

**Table 1 medicina-59-01026-t001:** Demographic and obstetric characteristics of term singleton breech deliveries.

	Epidural Analgesia N = 23	Remifentanil-PCA N = 104	*p* Value
Maternal age (years)	30.5 (3.9)	30.4 (4.8)	NS
Pre-pregnancy BMI (kg/m^2^)	22.9 (4.2)	22.4 (3.6)	NS
BMI at delivery (kg/m^2^)	27.9 (4.1)	27.3 (4.0)	NS
Nulliparity	17 (74%)	77 (74%)	NS
In vitro fertilization	2 (8.7%)	2 (1.9%)	NS
Spontaneous onset of labor	20 (87%)	84 (81%)	NS
Gestational age (weeks)	38.5 (1.0)	38.9 (1.1)	NS
Birthweight (g)	3157 (306)	3225 (415)	NS
SGA	2 (8.7%)	6 (5.8%)	NS
LGA	0	5 (4.8%)	NS

Note: Data are presented as mean (standard deviation) or n (%); Remifentanil-PCA: remifentanil patient-controlled analgesia; BMI: body mass index; SGA: small for gestational age (<10th centile according to national growth curves); LGA: large for gestational age (>10th centile according to national growth curves); NS: non-significant (*p* > 0.05).Table 2 presents the same comparison for twin deliveries at ≥34^+0^ weeks of gestation. The maternal body-mass index at delivery was significantly lower, and the proportion of nulliparous women was higher in the EA group. The birth weight of the first and second twins was statistically lower in the EA group.

Comparisons of the mode of delivery and adverse maternal as well as neonatal outcomes in EA vs. remifentanil-PCA in both breech and twin deliveries are presented in Table 3. There were no statistically significant nor clinically relevant differences between EA and remifentanil-PCA groups in terms of intrapartum CS rates, postpartum hemorrhage and OASI rates as well as in terms of low Apgar scores, birth asphyxia and NICU admission.

## 4. Discussion

In this single-center retrospective analysis of planned vaginal breech and twin deliveries, we found that type of labor analgesia (EA vs. remifentanil-PCA) was not associated with a higher risk of CS in labor, postpartum hemorrhage, OASI, low Apgar score, birth asphyxia or NICU admission in breech and twin deliveries. The findings of comparable safety and impact to the mode of delivery suggest that both analgesic options can be offered to women delivering twins and singleton term breech fetuses.

The results of our study are consistent with the recent Cochrane meta-analysis, which found similar rates of overall CS (RR 1.0, 95% CI 0.82–1.22) and low Apgar scores (RR 1.26, 95% CI 0.62–2.57) with EA compared to remifentanil-PCA [19]. A recent 5-year analysis of more than 10,000 deliveries at our center also showed no differences between EA and remifentanil-PCA in Apgar scores <7 at 5 min and NICU admissions among term deliveries of singletons in cephalic presentation [25]. However, in the same study, we found an association between remifentanil-PCA and lower CS rates compared to EA in nulliparous women with spontaneous and induced labor and in multiparous women with spontaneous onset of labor [25], which was not the case in the present analysis of breech and twin deliveries.

Our findings contrast with those of a retrospective single-center study by Chadha et al., which found a significantly higher CS rate with EA in the second stage of labor in women with a singleton breech presentation at term [39]. In this study, the increased likelihood of intrapartum CS in the second stage of labor was similar for both primiparous (odds ratio 5.43; 95% CI 2.46–1 1.95) and multiparous (odds ratio 5.37; 95% CI 2.07–13.87) parturients. However, while this likelihood was independent of the extent of cervical dilatation in the primiparae, it was only significant in the multiparae when initial cervical dilatation on admission was <3 cm (odds ratio 3.65; 95% CI 1.14–11.65) [39]. The contrast with our findings regarding intrapartum CS rates may be explained by the fact that we analyzed all CS in labor irrespective of the stage of labor. Chadha et al. only found an increased rate of CS associated with EA in the second stage but not in the first stage of labor.

Farghali et al. recently performed a prospective analysis of 343 parturients with twin gestation planned for a trial of vaginal delivery to determine the influence of EA on the delivery of the second twin [45]. The studied parturients received EA, while the control group consisted of parturients who received remifentanil-PCA on account of their ineligibility for EA. The authors concluded that compared to the remifentanil-PCA control, the use of EA decreases the incidence of CS for the delivery of both fetuses in twin gestation, as well as the frequency of combined CS and vaginal delivery for the birth of the second twin. Similarly, it was recently shown that in parturients with twin gestation attempting a trial of labor after a previous CS, using EA may decrease the risk of a repeat CS [46,47].

In 1977, Jaschevatzky et al. reported higher rates of operative vaginal deliveries and higher pre-term perinatal mortality in twin deliveries with EA, despite similar neonatal status (as assessed by the Apgar score at one minute) in both the EA and control groups [48]. Similarly, in a case series of parturients with multiple gestation who delivered vaginally, a higher incidence of low Apgar-minus-color scores at one minute among the second twins of at least 36 weeks gestation was reported in the EA group, although a shorter mean interval between delivery of the first and second twin was noted in the epidural series [49]. In contrast, and similar to our results, Weekes et al. found no association between EA and operative delivery, low Apgar scores, and perinatal mortality in twin deliveries [50]. In the same year, Gullestad and Sagen similarly reported no difference in neonatal outcomes of twins as assessed by Apgar score in parturients who received EA versus conventional analgesia [51]. It has to be noted that much more concentrated solutions of local anesthetics were used for EA in the 1970′s than those used today.

EA for vaginal breech delivery was reported to be associated with favorable Apgar scores and maternal outcomes and may be considered the preferred analgesic modality for this type of delivery [52,53]. Similar to our finding, Darby et al. reported no differences in emergency CS rates in patients with EA or parenteral analgesia in singleton breech deliveries [54]. However, contrary to our results, they reported that while the one-minute Apgar scores of the neonates in the EA and parenteral analgesia groups did not differ significantly, the Apgar scores of the neonates of primiparae were significantly higher in the EA group at five minutes. Our finding regarding neonatal outcome was consistent with another comparative study which found no difference with and without EA in singleton vaginal breech deliveries regarding mean 5 min Apgar scores and perinatal and maternal morbidity [55]. In contrast, a more recent study reported that EA was a significant risk factor for failed vaginal breech delivery and the consequent need for intrapartum CS [56].

Gowreesunker and Roelants reported two cases of remifentanil-PCA use for twin deliveries in parturients in whom EA was contraindicated due to anticoagulant therapy and coagulopathy [40]. It was observed that while delivery progressed uneventfully in one case, the delivery of the breech-presenting second twin in the second case required an urgent internal version and was associated with unsatisfactory efficacy of remifentanil-PCA. Accordingly, the authors suggested that remifentanil-PCA may not be preferred for twin deliveries because of the high risk of obstetrical maneuvers or CS for the birth of the second twin [40]. However, no evaluative study of the effects of remifentanil-PCA on labor outcomes in breech and twin deliveries has been published so far.

It should also be noted that other factors such as maternal age, nulliparity, and body-mass index may be independently associated with the observed increase in CS rate with EA in some studies [57]. In our study, for twin deliveries at ≥34^+0^ weeks of gestation, the proportion of nulliparous women was higher in the EA group. This is consistent with the findings of previous studies, which reported that EA was more frequently administered to lower-parity patients [58]. For twin deliveries, we found that the maternal body-mass index at delivery was significantly lower in the EA group. This contrasts with previous reports that suggest that obese parturients are more likely to receive neuraxial analgesia compared to those with a normal body-mass index [59,60]. Our analysis did not reveal any significant relationship between body-mass index and CS rate in the EA group, contrary to previous reports of a dramatic increase in caesarean delivery rate with increasing body-mass index among epidural recipients [61]. Similarly, our finding of lower birth weight in the first and second twins in the EA group also contrasts with other reports that suggest that the parturient’s requirement of EA is strongly related to a higher fetal birth weight [62]. However, the relatively small sample size in our study should be considered in interpreting these results.

The effects of EA and remifentanil-PCA on labor progress are highly dependent on the technique and dosages of medications used. Considering the substantial changes in labor analgesic techniques in the last few decades, more recent data are needed as the effects of EA and remifentanil-PCA on labor progress and outcomes are highly dependent on these factors. For example, in the Cochrane analysis of epidural versus alternative modalities of labor analgesia [7], while EA was associated with an overall increase in assisted vaginal delivery, a post hoc subgroup analysis revealed that this effect was absent in studies published after 2005, suggesting that more recent approaches to labor EA do not affect this outcome. A recent bibliometric analysis reported a marked increase in labor analgesia research between 1988–2018 [63]. However, as apparent in the preceding discussions, the available data on analgesic modalities or breech and twin deliveries are predominantly old retrospective studies and case reports. We believe that the results of the present study provide important updates that enhance the current understanding of maternal and fetal labor outcomes associated EA and remifentanil-PCA.

Our study had some limitations. First, the observational nature did not permit the controlling of all potential confounders (e.g., labor progress, anesthesia type for CS, etc.). As a result, it is not possible to draw any explicit conclusions on causality. Nevertheless, it is unlikely that a randomized trial on the effects of labor analgesia in twin or breech deliveries will be published in the next few years. Thus, the counseling of parturients on suitable analgesic methods in these labor categories would still be based on data from observational studies. Second, no data on neonatal cord pH or maternal hypoxemia were available in our dataset. While remifentanil-PCA may provide an alternative for labor analgesia in women who are not candidates for EA, caution is warranted particularly regarding hypoxemia. We have previously reported relatively high rates of maternal desaturation (34%), bradypnea (21%) and apnea (25%) associated with remifentanil-PCA at our center [18]. However, these events were transitional and easily managed. No serious respiratory depression or other serious complication occurred. This is in accordance with similar rates of neonatal birth asphyxia in the remifentanil-PCA and EA groups observed in the present study. Third, data were collected at a single tertiary perinatal center, which precludes generalization. We chose to analyze a 9-year period to avoid changes in clinical practices that may occur over more extended periods. This resulted in fairly small numbers of twin and breech deliveries. Although sample size calculation based on previously published data on intrapartum CS in breech and twin deliveries showed that the sample sizes in the present study could be sufficient, post hoc power analyses showed that the power of the tests performed in our retrospective study was actually <0.8 in several comparisons performed. Therefore, the study may not have been sufficiently powered to detect smaller differences in outcomes between the EA and remifentanil-PCA groups. Accordingly, further studies, including a larger pool of women with breech or twin deliveries, will be needed to confirm or refute our results.

## 5. Conclusions

Data on the safety of different analgesic methods in breech and twin deliveries are limited to a few older retrospective studies; hence, more recent studies are imperative to inform modern practice. It is unclear whether data on labor outcomes associated with EA and remifentanil-PCA can be generalized to all groups of parturients; to our knowledge, there have been no previously published evaluations of the impact of remifentanil-PCA on labor outcomes in breech and twin births. In the present study, we analyzed data obtained from the Slovenian National Perinatal Information System and found that type of labor analgesia (EA vs. remifentanil-PCA) was not associated with a higher risk of intrapartum CS, postpartum hemorrhage, OASI, an Apgar score of <7 at 5 min after birth, birth asphyxia or NICU admission in breech and twin deliveries. Our findings suggest that the current uses of EA and remifentanil-PCA are safe for pain relief in singleton breech and twin deliveries.

## Figures and Tables

**Table 2 medicina-59-01026-t002:** Demographic and obstetric characteristics of twin deliveries at ≥ 34^+0^ weeks of gestation.

	Epidural Analgesia N = 40	Remifentanil-PCA N = 204	*p* Value
Maternal age (years)	31.5 (4.7)	32.3 (5.0)	NS
Pre-pregnancy BMI (kg/m^2^)	22.6 (3.7)	23.8 (4.3)	NS
BMI at delivery (kg/m^2^)	27.7 (3.8)	29.6 (4.2)	0.01
Nulliparity	36 (90%)	144 (70.6%)	0.01
In vitro fertilization	17 (42.5%)	61 (29.9%)	NS
Spontaneous onset of labor	11 (27.5%)	72 (35.3%)	NS
Gestational age (weeks)	35.9 (1.04)	36.1 (1.2)	NS
First twin’s birthweight (g)	2404 (364)	2543 (352)	0.03
Second twin’s birthweight (g)	2277 (625)	2466 (401)	0.02
SGA second twin	9 (22.5%)	38 (18.6%)	NS
LGA second twin	0	1 (0.5%)	NS

Note: Data are presented as mean (standard deviation) or n (%); Remifentanil-PCA: remifentanil patient-controlled analgesia; BMI: body mass index; SGA: small for gestational age (<10th centile according to national growth curves); LGA: large for gestational age (>10th centile according to national growth curves); NS: non-significant (*p* > 0.05).

**Table 3 medicina-59-01026-t003:** Comparison of in-labor cesarean section (CS) rates, postpartum hemorrhage rates, obstetric anal sphincter injury (OASI) rates, low Apgar scores, birth asphyxia, and neonatal intensive care (NICU) admissions between epidural analgesia and remifentanil-PCA groups.

Term Breech Deliveries			
	Epidural Analgesia N = 23	Remifentanil-PCA N = 104	Odds Ratio (95% Confidence Interval) *
In-labor CS	10 (44%)	34 (33%)	0.75 (0.44–1.29)
Postpartum hemorrhage (>500 mL)	1 (4%)	9 (9%)	0.48 (0.06–3.99)
Postpartum hemorrhage (need for blood transfusion)	0	1 (1%)	NA
OASI	0	0	NA
Apgar score < 7 at 5 min	1 (4%)	3 (3%)	0.66 (0.07–6.09)
Mild to moderate birth asphyxia	2 (9%)	4 (4%)	2.38 (0.41–13.8)
Severe birth asphyxia	0	0	NA
NICU admission	3 (13%)	6 (6%)	0.44 (0.12–1.64)
Twin deliveries			
	Epidural analgesia N = 40	Remifentanil-PCA N = 204	Odds ratio (95% confidence interval) *
In-labor CS	5 (13%)	42 (21%)	1.65 (0.70–3.90)
Postpartum hemorrhage (>500 mL)	8 (20%)	42 (21%)	0.96 (0.41–2.25)
Postpartum hemorrhage (need for blood transfusion)	0	9 (4%)	NA
OASI	1 (3%)	1 (1%)	5.2 (0.32–85)
Apgar score <7 at 5 min (second twin)	1 (3%)	3 (2%)	0.59 (0.06–5.51)
Mild to moderate birth asphyxia (second twin)	1 (3%)	3 (2%)	1.72 (0.17–16.9)
Severe birth asphyxia (second twin)	0	0	NA
NICU admission (second twin)	7 (18%)	36 (18%)	1.00 (0.48–2.10)

Note: Remifentanil-PCA: remifentanil patient-controlled analgesia; * Odds ratios with 95% confidence intervals calculated for remifentanil-PCA; NA non applicable.

## Data Availability

The data reported in this study are available upon reasonable request to the corresponding author.

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
