# Peer review of "Intrapartum Cesarean Section and Perinatal Outcomes after Epidural Analgesia or Remifentanil-PCA in Breech and Twin Deliveries"

_medicina, 2023, doi:10.3390/medicina59061026_

Round 1

Reviewer 1 Report

Congratulation to the CS Rate of 21%

The introduction and the discussion are too long and should be shortened..

The Authors compared 23 cases vs 104 breach deliveries and 40 vs 204 twin deliveries. Statistical power? The authors should explain this discrepancy:! Is Remyfentanyl the standard procedure?

Quite important point is parity is the difference between primi- and multi-parous Women?

The CS Rate seems with 44% quite high, 33% in the ULTIVA group seems for me apropriate - The CI  of 0,44-1,29 shows that the data are not comparable.

The key message is that Ultiva can bei used instead of Epidural Analgesia - but the question with method has the better outcomes remains open.

They should add the pH -Values of the newborns!

Were there any instrumental deliveries in the Twin Group?

The methods of anaesthesiology are correct- maybe they should be evaluated by an anaesthesiologist 

Author Response

The introduction and the discussion are too long and should be shortened..

We have now tried to shorten the introduction and the discussion while still fulfilling the Journal’s 4000 words requirement.

The Authors compared 23 cases vs 104 breach deliveries and 40 vs 204 twin deliveries. Statistical power? The authors should explain this discrepancy:! Is Remyfentanyl the standard procedure?

Both EA and remifentanil were offered to women in labor with breech fetuses and twins at our institution during the study period. As stated in the manuscript, decision on labor analgesia method (remifentanil-PCA or EA) was taken by women during labor after consulting with anesthesiologists, obstetricians and midwives. Statistical power was assessed for both breeches and twins as reported in the Materials and methods section. We do agree that the numbers are small, especially for some sub-analyses. We have mentioned this among study limitations.

Quite important point is parity is the difference between primi- and multi-parous Women?

We agree with the reviewer. There was no significant difference in parity between EA and remifentanil-PCA groups in breech deliveries. However, there were more nulliparous women in the EA vs remifentanil-PCA group in twin deliveries. These comparisons are presented in tables 1 and 2 and also discussed in the Discussion section.

The CS Rate seems with 44% quite high, 33% in the ULTIVA group seems for me apropriate - The CI  of 0,44-1,29 shows that the data are not comparable.

CS rates for term breech deliveries in both EA and remifentanil groups reported are consistent with our institution CS rates in breech deliveries overall. The CI shows no statistically significant difference between in CS rates between groups.

The key message is that Ultiva can bei used instead of Epidural Analgesia - but the question with method has the better outcomes remains open.

Agree. The main finding of our study is that both analgesic methods have comparable safety profiles in the specific sub-populations of laboring women with breech fetuses and twins. We have emphasized this in the discussion and in the conclusions.

They should add the pH -Values of the newborns!

No data on cord pH are available in our database. This has been added to the discussion on limitations of the study.

Were there any instrumental deliveries in the Twin Group?

Forceps is not performed in breech deliveries at our institution. The Mariceau-Smellie-Veit maneuver is performed when needed for delivering the aftercoming head.

Reviewer 2 Report

Overall the introduction is too long and needs to be shortened considerably. Many of the cited references would be better as points in the discussion or deleted entirely.  

In the materials and methods, you state that there were 5000 - 5500 deliveries per year over the 9 years which would be looking back at 45,000 - 50,000 deliveries. Then when I look at the numbers of patient's evaluated in the study it is only 23 with epidurals compared to 104 with PCA pumps and for the twins it is only 40 with epidurals compared to 204 with PCA pumps. This is an incredibility small sample size to determine if the outcomes are superior in one group compared to another. Did you consider doing this study as a non-inferiority trial? Furthermore if you then look at your sub comparisons of outcomes for the term breech you had 0 patients in the epidural group that needed a blood transfusion, 1 that had an Apgar score of < 7 at 5 minutes and 0 with severe birth asphyxia. In the twin deliveries you also had 0 who needed a blood transfusion, and 0 had severe birth asyphyxia.  Did patients either get offered epidural or PCA. What percentage of the women who delivered in your hospital got epidurals and how many used PCA?  I would assume that most chose one of the other. If so then how did you select the 23 women who had epidurals and had a term breech that you would analyze? Or the 40 women who had epidurals to compare to the women who you selected for the PCA group for the comparison.  You also used blood loss of > 500 ml as postpartum hemorrhage.  And if the women self selected then how can you be sure the groups are similar if the patients self selected.  Most Societies use > 1000 ml as threshold for PPH, because loss of 500 cc is clinically insignificant without a change in pulse or alteration in blood pressure. 

English is acceptable.

Author Response

Overall the introduction is too long and needs to be shortened considerably. Many of the cited references would be better as points in the discussion or deleted entirely. 

Agree. We have now tried to shorten the introduction and the discussion while still fulfilling the Journal’s 4000 words requirement.

In the materials and methods, you state that there were 5000 - 5500 deliveries per year over the 9 years which would be looking back at 45,000 - 50,000 deliveries. Then when I look at the numbers of patient's evaluated in the study it is only 23 with epidurals compared to 104 with PCA pumps and for the twins it is only 40 with epidurals compared to 204 with PCA pumps. This is an incredibility small sample size to determine if the outcomes are superior in one group compared to another. Did you consider doing this study as a non-inferiority trial? Furthermore if you then look at your sub comparisons of outcomes for the term breech you had 0 patients in the epidural group that needed a blood transfusion, 1 that had an Apgar score of < 7 at 5 minutes and 0 with severe birth asphyxia. In the twin deliveries you also had 0 who needed a blood transfusion, and 0 had severe birth asyphyxia. 

This was a retrospective analysis of all labors with singleton breech fetuses and twins without contraindications for trial of vaginal delivery at our institution during the study period. We agree the numbers are small and have emphasized this in the discussion section. We have also emphasized that the statistical power assessed for CS rates analysis cannot be generalized to all sub-analyses performed.

Did patients either get offered epidural or PCA.

Both EA and remifentanil were offered to women in labor with term breech fetuses and twins at our institution during the study period, i.e. remifentanil was not te standard procedure. As stated in the manuscript, decision on labor analgesia method (remifentanil-PCA or EA) was taken by women during labor after consulting with anesthesiologists, obstetricians and midwives.

What percentage of the women who delivered in your hospital got epidurals and how many used PCA? I would assume that most chose one of the other. If so then how did you select the 23 women who had epidurals and had a term breech that you would analyze? Or the 40 women who had epidurals to compare to the women who you selected for the PCA group for the comparison. 

The overall rate of EA during the study period was 35% and 25%for remifentanil. Our analysis showed that more women with breech fetuses or twins opted for remifentanil labor analgesia. All breech and twin deliveries with either EA or remifentanil during the study period were included.

You also used blood loss of > 500 ml as postpartum hemorrhage.  Most Societies use > 1000 ml as threshold for PPH, because loss of 500 cc is clinically insignificant without a change in pulse or alteration in blood pressure. 

Agree. Our database registers PPH of >500mL. This is why we also added the analysis RBC transfusions.

And if the women self selected then how can you be sure the groups are similar if the patients self selected. 

This was a retrospective cohort study. An attempt to compare important obstetric characteristics was made (tables 1 and 2), but the observational nature of the study does not allow to control for all potential confounders. This is discussed among the study limitations. However, it is very unlikely that a large RCT on labor analgesia in breech or twin deliveries will be performed soon. Therefore, we will need to use observational data to assess safety of labor analgesia methods in these high-risk deliveries.